# Ensuring that COVID-19 research is inclusive: guidance from the NIHR INCLUDE project

Miles D Witham ![ORCID] ,[1,2] Eleanor Anderson,[2] Camille B Carroll ![ORCID] ,[3] Paul M Dark,[4] Kim Down,[2] Alistair S Hall,[5] Joanna Knee,[6] Eamonn R Maher,[7] Rebecca H Maier,[8,9] Gail A Mountain,[10] Gary Nestor,[2] John T O'Brien,[11] Laurie Oliva,[12] James Wason,[9,13] Lynn Rochester,[2,14] On behalf of the NIHR CRN INCLUDE Steering Group

For numbered affiliations see end of article.

**Correspondence to**
Professor Miles D Witham;
miles.witham@newcastle.ac.uk

## ABSTRACT

**Objective** To provide guidance to researchers, funders, regulators and study delivery teams to ensure that research on COVID-19 is inclusive, particularly of groups disproportionately affected by COVID-19 and who may have been historically under-served by research.

**Summary of key points** Groups who are disproportionately affected by COVID-19 include (but are not limited to) older people, people with multiple long-term conditions, people with disabilities, people from Black, Asian and Ethnic minority groups, people living with obesity, people who are socioeconomically deprived and people living in care homes. All these groups are under-served by clinical research, and there is an urgent need to rectify this if COVID-19 research is to deliver relevant evidence for these groups who are most in need. We provide a framework and checklists for addressing key issues when designing and delivering inclusive COVID-19 research, based on the National Institute for Health Research INnovations in CLinical trial design and delivery for the UnDEr-served project roadmap. Strong community engagement, codevelopment and prioritisation of research questions and interventions are essential. Under-served groups should be represented on funding panels and ethics committees, who should insist on the removal of barriers to participation. Exclusion criteria should be kept to a minimum; intervention delivery and outcome measurement should be simple, flexible and tailored to the needs of different groups, and local advice on the best way to reach and engage with under-served communities should be taken by study delivery teams. Data on characteristics that allow identification of under-served groups must be collected, analyses should include these data to enable subgroup comparisons and results should be shared with under-served groups at an early stage.

**Conclusion** Inclusive COVID-19 research is a necessity, not a luxury, if research is to benefit all the communities it seeks to serve. It requires close engagement with under-served groups and attention to aspects of study topic, design, delivery, analysis and dissemination across the research life cycle.

## Strengths and limitations of this study

► We provide detailed guidance across the research life cycle—a whole-systems approach to improve inclusion in COVID-19 research.
► We base the guidance on the robustly developed INnovations in CLinical trial design and delivery for the UnDEr-served roadmap for improving inclusion in research.
► The novelty of COVID-19 research precludes our recommendations being based on evidence directly derived from studies with people with COVID-19.

research response.[1] Across the world, teams are designing and delivering studies encompassing a range of COVID-19 research including epidemiology, disease surveillance, disease consequences, early and late phase clinical trials. Early evidence suggests that some segments of the population are at high risk either of contracting COVID-19 or of more severe consequences of COVID-19 including hospitalisation and death.[2 3] Some of these groups have not been well reached by traditional research design and delivery mechanisms,[4] and there is growing concern that some of the groups most vulnerable to the impact of COVID-19 are under-represented in research studies.[5]

This guidance is designed to facilitate best practice in the design, funding, approval, regulation and delivery of research on COVID-19 and is a product of the ongoing National Institute for Health Research (NIHR) INnovations in CLinical trial design and delivery for the UnDEr-served (INCLUDE) project which seeks to improve research for under-served groups; the guidance has been made available by the NIHR website.[6]

### What is an under-served group?
There is no single definition of an under-served group in research; the definition is

## INTRODUCTION
COVID-19 is challenging health systems globally and has triggered an unprecedented

context-dependent and will vary depending on the population, disease and intervention being studied.[7] However, if a group is enrolled in a study at lower rates than one sees in the population affected by the disease, this is evidence that the group is likely to be under-served. Reasons for a group being under-served may include protocol design exclusions, an intervention that is unsuitable for that group or because research delivery does not enable them to participate in practice.

### Who is at risk of contracting COVID-19?

Our knowledge of who is more or less likely to be infected with COVID-19 (as opposed to suffering severe consequences of infection) is limited at present. However, individuals with a high degree of contact with those who are infected, including healthcare workers, retail staff, transport staff and other key workers with face-to-face roles, are likely to be at high risk. Other groups in which risk appears elevated are patients who spend prolonged periods of time in close proximity to other patients (eg, patients attending hospital-based haemodialysis[8]) and those in care homes[9]; emerging evidence suggests that once COVID-19 starts to affect a care home, it can spread very rapidly through the care home population of both residents and staff.

### Who is at risk of severe consequences of COVID-19?

Emerging data suggest that several groups are at a higher risk of death or critical care unit admission from COVID-19 infection.[2 3 10] These groups include, but are not limited to, older people (particularly those aged 70 and over); men; people living with obesity; people with disabilities, people living with multiple long-term conditions; care home residents and others living with frailty; Black, Asian and minority ethnic (BAME) groups; and people with immunosuppression due to medication, systemic illness or malnutrition.

### What groups may be under-served by healthcare and research systems with respect to COVID-19?

This is also likely to be context-specific, but examples include the following.
► Older people and others who are self-isolating (who may not be able to access healthcare easily) on government advice.
► Those who may not be prioritised for hospital admission or critical care unit admission due to perceived limited life expectancy or a low chance of benefitting from such interventions.
► Those who lack digital literacy or access to digital technologies.
► People with cognitive problems including dementia, especially those living alone, who may not be able to understand and/or adhere to the symptom reporting and testing needed for diagnosis.
► People with disabilities, who may not be able to access information or testing facilities and may struggle to access healthcare and care support during periods of lockdown.[11]
► Socioeconomically deprived groups, who are likely to have less resilience to the economic shocks precipitated by the COVID-19 pandemic response.
► People without a car or who live in rural areas (who may find it difficult to access testing facilities that require attendees to come in a car).

In addition, thought needs to be given to the fear that many of the population currently have of leaving their immediate environment, entering a hospital or other healthcare facility. This may disproportionately affect certain segments of the population (eg, those with anxiety disorders or who perceive themselves to be especially vulnerable).

### How can under-served groups be more successfully included in COVID-19 research?

The INCLUDE project contains a roadmap outlining the potential points of intervention to improve the inclusion

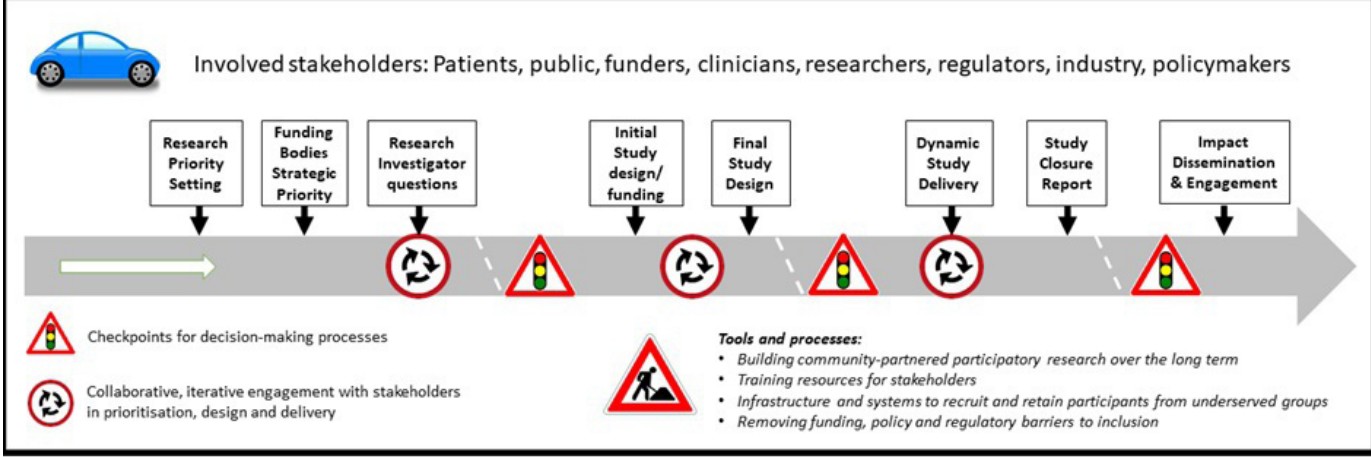

**Figure 1** The INCLUDE roadmap (reproduced from National Institute for Health Research [6] under a creative commons attribution 4.0 international license).

of under-served groups (figure 1) and a structure that may be helpful in guiding teams considering how to focus their efforts. Ensuring that effective training for researchers, funders, delivery teams and regulators is key, as is ensuring that regulation is proportionate and does not place barriers in the way of including under-served groups. Identifying members of under-served groups and using their knowledge is critical to successful delivery of research, as is building on existing resources that have been created to facilitate research for specific under-served groups such as older people[12] or people from BAME communities.[13]

### Recommendations from the include project for COVID-19 research

Below, we set out a series of recommendations across the research life cycle for how under-served groups can be included more successfully in COVID-19 research.

#### Research priority setting

► Representatives from under-served groups should be consulted at the earliest possible time to agree a set of national research priorities for COVID-19 research. A rapid James Lind Alliance project (a priority-setting process co-produced by researchers, patients and a range of other stakeholders) would be one possible way to achieve this, but opportunities also exist to ensure representation on COVID-19 funding committees and prioritisation committees.

#### Strategic priorities for funding bodies

► Strategic priorities for research funding bodies (eg, government funders and charities) should reflect the views of under-served groups from any national or international research priority exercise, as well as reflecting current scientific paradigms and national health and social care system imperatives.
► Patient and public representation should be in place on all COVID-19-related funding and prioritisation panels, and this patient and public representation should encompass representatives from groups who are at risk of being under-served by COVID-19 research.

#### Research investigator questions

► Specific research questions to be answered should be codeveloped by research teams, funders, government and patient and public representatives including people from under-served groups.

#### Initial design and funding

► Inclusion criteria for research studies should be kept to the minimum necessary to ensure patient safety and criteria likely to exclude broad sectors of populations should be avoided. In particular, we recommend that studies do not overtly or inadvertently exclude participants on the basis of age, sex, ethnicity, geography, socioeconomic status, multimorbidity, disability, cognitive impairment, body mass index or place

of residence unless studies specifically aim to generate data for groups that have to date been under-served by COVID-19 research.
► Interventions should be designed and selected to be usable by a wide range of participants, including those from under-served groups. If this is not possible, then serious consideration should be given to abandoning the intervention on the grounds that it will not be broadly applicable in practice. Exceptions to this are interventions designed to target a specific biological pathway identified by disease severity or other disease characteristics (ie, stratified or precision medicine approaches), or interventions designed specifically to target under-served groups where current approaches are not sufficiently effective.
► Outcomes should be selected to be relevant to a broad range of participants; outcomes should be simple and should place minimal burden on both participants and research teams. Outcomes should be deliverable under pandemic conditions (eg, when wearing full personal protective equipment (PPE) in overloaded hospitals). Outcomes that can be collected from routinely collected clinical data or participant-reported outcomes that have the flexibility to be collected by telephone, by video link, by remote devices (eg, accelerometers or other Bluetooth-enabled devices[14]) or by web-based/SMS interfaces should be preferred, ensuring that such platforms can be used by participants with a range of sensory and motor disabilities.
► When outcomes must be collected face to face, these should be able to be collected in a range of different environments including the participant's own home, not just in hospital wards or clinics. Outcomes should, where possible, be able to be assessed by a range of staff with no or minimal training to ensure that studies are resilient against staff illness or redeployment in future pandemic waves. Research teams should build the costs of hardware, software, accommodations for people with disability and adequate time to support these outcome collection methods into their proposals.
► Researchers should design study protocols to allow recruitment through as wide a range of routes as possible. Without good scientific reason, recruitment should not be limited to hospitals, primary care practices or care institutions, but should be enabled through all of these routes and others including through community social hubs and organisations, via adverts, web-based signup, verbally over the telephone, in response to SMS and traditional mail, and other routes appropriate to reach particular under-served groups. The emergence of new community networks to support isolated and vulnerable people during the pandemic provides new opportunities for contacting and engaging under-served groups in research. Key issues to consider when designing COVID-19 studies are given in box 1.

---

**Box 1  A checklist to guide researchers in designing inclusive COVID-19 research**

► Have I engaged with representatives of under-served groups in designing my study question and protocol?
► Have I avoided inclusion/exclusion criteria likely to lead to exclusion of under-served groups? (eg, age limits, excluding multimorbidities, body mass index limits)
► Is my study intervention suitable for, and acceptable to, under-served groups? (eg, avoiding gelatine capsules on medication)
► Are my outcomes validated and relevant to a broad range of patients in the populations that my research seeks to serve? (eg, cognition assessment tools in translated versions)
► Have I avoided study burden that is likely to make it more difficult for some groups to participate? (eg, frequent study visits)
► Have I provided a range of recruitment pathways that give flexibility in how participants are sought and contacted? (eg, not confining recruitment to drive-in testing centres, allowing recruitment from care homes)
► Have I provided study information in a format that is accessible to under-served groups? (eg, translations of study information, simplified study information, information written with a reading age of 9 or below, written in large font size, screen reader compatible)
► Have I enabled consent to be obtained in a way that is flexible and tailored to the needs of different groups? (eg, EConsent for those isolating, assent or supported decision making from relatives for those with cognitive impairment). Have I involved carers and assentees in the design of the assent process for studies where consent cannot be obtained from the participant?
► Will my study still be deliverable in the event of a second pandemic wave, with a possible increase in lockdown restrictions and study staff redeployment?

---

**Box 2  A checklist to guide funders and reviewers in assessing inclusiveness of COVID-19 research**

► Does the topic of the study reflect the priorities elicited by consultation with under-served groups?
► Have representatives from under-served groups been engaged in the design of the study, and in what way?
► Does the target population for the study (as defined by the inclusion and exclusion criteria) reflect those who are at risk of contracting COVID-19 or who are at increased risk of the adverse consequences of COVID-19?
► Have the investigators taken steps to ensure that these potentially under-served groups will be included in the study with appropriate mitigation of any risks?
► Are there unnecessary or unjustifiable exclusion criteria (eg, old age, sex, ethnicity, obesity, cognitive impairment, multimorbidity) or research methods that will act as barriers to inclusion of under-served groups at high risk of COVID-19 infection and its adverse consequences?
► Is the intervention designed and delivered in a way that is acceptable and feasible to a broad range of people most at risk from COVID-19 infection?
► Are the study outcomes easy to measure, and can they be measured in a range of different environments? Will they still be able to be measured during the peak of a pandemic wave or during population lockdown?
► Does the study target a specific under-served group? If so, is adequate justification for this given, and is the strategy proposed to target the group likely to succeed?

---

► Funders can apply the checklist given later in this document (see box 2) to assess whether research proposals have been developed in conjunction with under-served groups and that proposals are designed and will be delivered in a way likely to be able to recruit under-served groups successfully in COVID-19 research.
► Funders should ensure that additional funds (above and beyond the amount of money traditionally viewed as adequate to support recruitment and retention) are available to research teams to support the successful recruitment of under-served groups. This may require funds for home visiting, transportation (in many cases by car or taxi, not public transport), PPE, more screening per participant recruited, longer study visits, provision of translators or translated materials.

### Final study design

► Research Ethics Committees should pay particular attention to inclusion and exclusion criteria, routes of recruitment, information materials, consent processes (including the needs of those without capacity) and whether under-served groups have been engaged in study design. Where study designs seem likely to fail to include sufficient people from under-served groups, ethics committees should robustly call attention to

this and ask research teams to modify their proposals accordingly even if doing so leads to short delays in approval of projects. Key issues to consider are listed in box 2.
► Sponsor organisations should apply similar scrutiny to research proposals at the design stage and insist on changes where the design makes the exclusion of under-served groups likely.
► Sponsor organisations and Research Ethics Committees should both welcome and encourage novel and efficient methods of approaching, consenting, recruiting and retaining participants in research studies. Many proposals will be very different from those that such organisations are used to seeing, but this should be viewed as an opportunity, not a reason to fall back on traditional (and often ineffective) methods of study conduct.

### Study delivery

► Local research delivery teams should seek local advice on how to reach and engage with under-served groups in their areas. In many cases, this will entail getting advice from members of specific ethnic communities, or those with lived experience of particular population sectors (eg, those living in care homes, homeless people). In selected cases, this may extend to using members of under-served communities to deliver research. Key issues to consider are listed in box 3.

> **Box 3    A checklist to guide research delivery teams in delivering inclusive COVID-19 research**
>
> ► Have we engaged with local representatives from under-served groups to understand how best to approach, recruit and retain participants to this study locally?
> ► Are our staff trained to approach and recruit people from under-served groups? (eg, those living in care homes). If not, what training is required (eg, age inclusion training, disability inclusion training) and how will we deliver this?
> ► Do our staff have the correct equipment to deliver the study to under-served groups (eg, portable testing and blood taking kit, car transport and PPE to conduce home visits, contracts and lone working policies to support home working, video and phone consulting systems to enable remote follow-up, accommodations to support the inclusion of people with disabilities)?
> ► Do we have staff with particular skills or background who would be well suited to engage with under-served groups (eg, speak the same language, live in the same area, experience working in a particular healthcare sector)?
> ► Do we have processes in place locally to monitor whether we are reaching people from under-served groups—and if not, why not?
> ► If recruiting people from under-served groups is taking more time or resources, is the extra time and resource being made available, either from the funder or from local resources?

► Training for research delivery staff (local investigators, research nurses and other research team members) should be delivered to raise awareness of the need to recruit inclusively, to highlight particular under-served groups and to provide the generic skills required to engage with specific under-served groups.

► Local research teams should configure themselves to be able to recruit in environments other than standard inpatient and outpatient secondary care departments. This should include conducting study procedures in participants' own homes, care homes, conducting study procedures remotely (eg, via telephone and video links) and taking care to accommodate the needs of those with a range of disabilities. Linking with neighbouring healthcare research organisations will enable cross-cover of research staff in the event of staff sickness or of one organisation becoming overloaded by clinical work in future pandemic waves.

► For trials of medicinal products, dispensing arrangements for oral medications should be put in place to enable the delivery of medications directly to participants' homes or local pharmacy rather than needing to collect medication from a research facility. This approach will help to build resilience against future restrictions on movement or closure of workplaces.

► Local research support agencies (such as the UK NIHR local Clinical Research Networks and hospital Trusts) should provide local research teams with the equipment and resources that they need to deliver research in non-traditional settings. This may include transportation, contracts and working practice procedures enabling work in participants' homes, provision of COVID-19 testing and PPE in line with national guidance, IT hardware and software to enable video links, EConsent and electronic case record form use while roaming, translation services and sufficient time to conduct study visits in non-traditional settings.

► Local research support agencies should put in place systems to monitor local recruitment of under-served groups to each study. Combining these local data across studies will enable study teams to adapt their recruitment strategies at an early point, for instance, switching place of recruitment, oversampling particular under-served groups or changing how study teams approach participants.

### Study closure and analysis

► Investigators and sponsors should develop plans for how pauses or discontinuation of study activity due to future pandemic waves will be managed (ideally to at least enable the study follow-up to continue), and how these plans will be communicated to participants.

► Study reports and analysis plans should report the proportion of relevant under-served groups and compare these proportions to those found in the general population with COVID-19 illness.

► Study analysis plans should include adequately powered prespecified subgroup analyses for key under-served groups, both to explore differences within studies and to enable later pooling of results from different under-served groups.

### Impact, dissemination and engagement

► All studies should devise a comprehensive dissemination and engagement plan, and representatives from under-served groups should be part of the team drawing up these plans.

► Specific engagement plans for different under-served groups should be drawn up to enable appropriate tailoring of messages to different groups, in a way that best encourages feedback, debate and engagement within different groups.

### CONCLUSION

At a time when research on COVID-19 is being designed and delivered at an extraordinary speed, it may seem that the urgency of the situation obviates the need to fully engage under-served groups in the design and delivery of COVID-19 research. We argue however that doing so is not a luxury but a necessity and failure to engage represents an enormous wasted opportunity.[15] It will hamper scientific progress and potentially lead to the deployment of ineffective or harmful diagnostics and therapeutics to large sections of the population. This in turn will lead to worse health outcomes for under-served groups, and healthcare resources being wasted. Fully engaging under-served groups and enabling their inclusion in COVID-19 research is the only way that we will be able to understand the health impacts of COVID-19 in these most vulnerable groups and thus mitigate the impact of COVID-19 on health and society in an equitable way.

**Author affiliations**
[1]NIHR Newcastle Biomedical Research Centre, Newcastle University, Newcastle upon Tyne, UK
[2]NIHR Clinical Research Network Cluster E, Newcastle University, Newcastle upon Tyne, UK
[3]Faculty of Health, University of Plymouth, Plymouth, UK
[4]NIHR Manchester Biomedical Research Centre, The University of Manchester and Northern Care Alliance NHS Group, Manchester, UK
[5]Cardiology Department, Leeds General Infirmary Department of Cardiology, Leeds, UK
[6]NIHR Clinical Research Network Coordinating Centre, University of Leeds, Leeds, UK
[7]Department of Medical Genetics, University of Cambridge and NIHR Cambridge Biomedical Research Centre, Cambridge, UK
[8]Newcastle Clinical Trials Unit, Newcastle University, Newcastle upon Tyne, UK
[9]Population Health Sciences Institute, Newcastle University, Newcastle upon Tyne, UK
[10]Centre for Applied Dementia Studies, University of Bradford, Bradford, UK
[11]Department of Psychiatry, University of Cambridge, Cambridge, UK
[12]NIHR Clinical Research Network Coordinating Centre, Guy's and St Thomas' NHS Foundation Trust, London, UK
[13]MRC Biostatistics Unit, University of Cambridge, Cambridge, UK
[14]Brain and Movement Group, Translational Clinical Research Institute, Newcastle University, Newcastle upon Tyne, UK

**Contributors** MW: conceptualisation, wrote first draft, critical revision and editing, approval of final version. EA, CBC, PD, KD, ASH, JK, EM, RM, GM, GN, JO, LO, JW and LR: conceptualisation, critical revision and editing, approval of final version.

**Funding** This work was undertaken as part of the INCLUDE project, which is commissioned and funded by the UK National Institute for Health Research Clinical Research Network Coordinating Centre.

**Competing interests** None declared.

**Patient consent for publication** Not required.

**Provenance and peer review** Not commissioned; externally peer reviewed.

**ORCID iDs**
Miles D Witham http://orcid.org/0000-0002-1967-0990
Camille B Carroll http://orcid.org/0000-0001-7472-953X

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
