## [Reviewer comments · BMJ Open]

ARTICLE DETAILS

TITLE (PROVISIONAL)	Ensuring that COVID-19 Research is Inclusive - guidance from the NIHR INCLUDE project
AUTHORS	Witham, Miles; Anderson, Eleanor; Carroll, Camille; Dark, Paul; Down, Kim; Hall, Alistair; Knee, Joanna; Maher, Eamonn; Maier, Rebecca; Mountain, Gail; Nestor, Gary; O'Brien, John; Oliva, Laurie; Wason, James; Rochester, Lynn

VERSION 1 – REVIEW

REVIEWER	Dikaios Sakellariou Cardiff University, UK
REVIEW RETURNED	16-Sep-2020

GENERAL COMMENTS	Dear Authors, This communication article addresses a very important and topical issue; as you state in the abstract, inclusive COVID-19 research is a necessity, and therefore clear guidelines are needed. This short piece sets a clear framework which can guide funding bodies and researchers. The text is clearly written and clearly communicates your arguments. My only suggested change would be to clearly mention people with disabilities as one of the groups who are disproportionately affected by the COVID-19 pandemic and by the measures taken to address it. While you mention people with chronic conditions, this does not necessarily cover people with disabilities, who may face specific issues (e.g. lack of accessible public health information, lack of accessible testing facilities etc.). I look forward to seeing this manuscript in print.
--

REVIEWER	Morgon Banks London School of Hygiene & Tropical Medicine
REVIEW RETURNED	29-Sep-2020

GENERAL COMMENTS	I think this paper is well-written and covers an important topic. My only major critique is that there is no mention of people with disabilities, who are a critically under-served group during the current pandemic. For example, the ONS found that mortality rates from COVID-19 were 2.4x higher for men with disabilities and 1.9x higher for women with disabilities in England and Wales compared to their counterparts without disabilities. People with disabilities are also at risk of exclusion from COVID-19 prevention and response measures, as well as research on COVID-19, due for example to lack of communication/information in accessible formats for people with sensory and intellectual impairments (including consent, recruitment and data collection procedures), physically inaccessible infrastructure/equipment, misconceptions and stigma
---

	of disability and lack of disability-specific guidance and training (e.g. preventative measures for people requiring personal assistance, researcher training on working with people with different impairment types). See for example, Kuper et al 2020 (Disability-inclusive COVID-19 response: What it is, why it is important and what we can learn from the United Kingdom's response). While there is some overlap between people with disabilities and other groups mentioned in this paper, I do think an explicit mention of disability is needed.
--	---

VERSION 1 – AUTHOR RESPONSE

Reviewer 1: "clearly mention people with disabilities as one of the groups who are disproportionately affected by the COVID-19 pandemic and by the measures taken to address it. "

We have now:

- Included mention of people with disabilities in those at risk of severe COVID (p6 para 1), and those disproportionately affected by the pandemic (p6 para 2)
- Included disabilities in the list of exclusion criteria to be avoided (p7 para 1)
- Included mention of people with disabilities in the abstract (p3 para 2)

Reviewer 2: "My only major critique is that there is no mention of people with disabilities, who are a critically under-served group during the current pandemic."

We have now:

- Included mention of people with disabilities in those at risk of severe COVID (p6 para 1), and those disproportionately affected by the pandemic (p6 para 2)
- Included disabilities in the list of exclusion criteria to be avoided (p7 para 1)
- Included mention of people with disabilities in the abstract (p3 para 2)
- We have also included the suggested additional reference (now reference 10) (p6 para 2)

We hope that these additions satisfactorily reflect the importance of disability correctly highlighted by the reviewers

VERSION 2 – REVIEW

REVIEWER	Dikaios Sakellariou Cardiff University, UK
REVIEW RETURNED	10-Oct-2020

GENERAL COMMENTS	I am satisfied with the changes made. i look forward to seeing this manuscript in print.
--

REVIEWER	Morgon Banks London School of Hygiene & Tropical Medicine, UK
REVIEW RETURNED	12-Oct-2020

GENERAL COMMENTS	Just a few minor points: - Last paragraph of page 8, "Research teams should build in the costs of hardware, software and adequate time to support these
--

	outcome collection methods into their proposals." Would also include accommodations for people with disabilities.  - Similarly for second to last paragraph on page 8 (on remote data collection), would add that should use platforms that are accessible to people with sensory disabilities (e.g. screen reader compatible, closed captioning). This point should also be emphasised in the 3rd paragraph of "study delivery" - 3rd paragraph under "study closure and analysis": I think worth adding a line about ensuring the sample size is adequate for sufficiently powered for these sub-group analyses - Pg 10: "Local research delivery teams should seek local advice..." what about inclusion in the research team where possible? - Box 1, pt 7: would add a few more disability specific references, such as screen reader compatible - Box 1, pt 8: "assent from relatives for those with cognitive impairment" - would be careful with this, typically supported decision-making is what's recommended - Box 3, pt 2: would add disability-inclusion training - Box 3, pt 3: would add "accommodations to support the inclusion of people with disabilities"
--	--

VERSION 2 – AUTHOR RESPONSE

Reviewer 1: No comments

Reviewer 2:

Last paragraph of page 8, "Research teams should build in the costs of hardware, software and adequate time to support these outcome collection methods into their proposals." Would also include accommodations for people with disabilities.

We have added the suggested phrase at the end of page 8

- Similarly for second to last paragraph on page 8 (on remote data collection), would add that should use platforms that are accessible to people with sensory disabilities (e.g. screen reader compatible, closed captioning).

We have added a line about this at the end of the paragraph (p8 para 3)

- This point should also be emphasised in the 3rd paragraph of "study delivery"

We have added an additional line (p10 para 5) emphasising this point as requested

- 3rd paragraph under "study closure and analysis": I think worth adding a line about ensuring the sample size is adequate for sufficiently powered for these sub-group analyses

We have added 'adequately-powered' to the point about prespecified subgroup analyses in this paragraph now (p11 para 5)

- Pg 10: "Local research delivery teams should seek local advice..." what about inclusion in the research team where possible?

We had added a line now (p10 para 3) suggesting using members of under-served communities to deliver research

- Box 1, pt 7: would add a few more disability specific references, such as screen reader compatible

We have added this suggestion

- Box 1, pt 8: "assent from relatives for those with cognitive impairment" - would be careful with this, typically supported decision-making is what's recommended

We have added supported decision-making to this point, although this applies only to those with sufficient capacity to take part in participation decisions

- Box 3, pt 2: would add disability-inclusion training

We have added this and another example to this point
- Box 3, pt 3: would add "accommodations to support the inclusion of people with disabilities"
We have added the suggested line to this point

We hope that these changes satisfactorily address the points raised. Although there is always more to say about any particular under-served group, we feel that there is a risk that further emphasis on people with disability would unbalance the article with undue emphasis on this under-served group in comparison with other under-served groups.

VERSION 3 – REVIEW

REVIEWER	Morgon Banks London School of Hygiene & Tropical Medicine
REVIEW RETURNED	19-Oct-2020
GENERAL COMMENTS	Paper looks great, look forward to seeing it in print